# Quality and Biodegradation Process of Dissolved Organic Carbon in Typical Fresh-Leaf Leachate in the Wuhan Urban Forest Park

Xiaokang Tian and Siyue Li *

School of Environmental Ecology and Biological Engineering, Institute of Changjiang Water Environment and Ecological Security, Hubei Key Laboratory of Novel Reactor and Green Chemical Technology, Wuhan Institute of Technology, Wuhan 430205, China; 1517639207@qq.com
* Correspondence: syli2006@163.com

**Abstract:** The study investigated the leaching and biodegradation of dissolved organic carbon in leaf leachates from typical fresh leaves in the Wuhan Urban Forest Park, Central China. The fresh leaf-leached dissolved organic carbon quality and biodegradability, as well as their potential determinants, were examined for 12 major tree species, including deciduous trees and shrubs. A 28-day indoor incubation was conducted at two temperature conditions of 20 °C and 30 °C. Sampling was conducted within the planned time frame for experimental measurements, and a first-order kinetic model for dissolved organic carbon degradation was fitted. The utilization of the fir tree as the predominant deciduous species and cuckoo as the primary shrubs provided advantages in increasing the carbon sequestration capacity of urban forests. There was no significant difference in the degradation rate of the leaching solution at different temperatures, but the k value of the first-order kinetic model was different. At 20 °C, the dissolved organic carbon degradation rate was positively correlated with electrical conductivity and total dissolved nitrogen, while it was negatively correlated with the humification index and ratio of dissolved organic carbon to total dissolved nitrogen. At 30 °C, the degradation rate of dissolved organic carbon showed a positive correlation with total dissolved phosphorus and total dissolved nitrogen, while it was negatively correlated with the humification index, ratio of dissolved organic carbon to total dissolved nitrogen and ratio of dissolved organic carbon to total dissolved phosphorus.

**Keywords:** leachate; incubation; DOC degradation; kinetic model; influencing factors

## 1. Introduction

The plant litter in urban forest parks was considered to be an important non-point source of carbon and nutrient substances for urban water bodies [1,2]. The leachates from leaves and litter carry substantial dissolved organic matter, infiltrating the soil environment and participating in the forest's nutrient cycling [3]. The study of leaves became increasingly important for understanding the input of organic matter in ecosystems. However, less attention was given to the topic.

Until now, researchers have studied the DOM (dissolved organic matter) (all abbreviations and symbols in this article could be found in Table S1) leached from trees. Regarding the whereabouts of litter DOC (dissolved organic carbon), it was confirmed that most of it remained in the soil environment, with a small part being mineralized [4]. There were significant differences in the production and degradation of DOC from different plant litter sources, and the structural and physiological changes in plants also had significant effects on DOC production [5]. Under the condition of global warming, Hagedorn et al. found that, with the increase in atmospheric $CO_2$, the initial DOC leaching of forest litter increased, while the biodegradable part decreased [6]. Additionally, the initial carbon quality of plant litter also affected the decomposition and carbon sequestration of litter [7]. Regarding

studying the leaching and degradation of litter, Xu et al. [8] found that the DOM leaching amount in leaves was more than that in other organs, and the biodegradability of DOM leaching was low, which was affected by DOM aromatics and nitrogen (N) availability. Others demonstrated that the composition and molecular weight of DOC could also impact microbial decomposition and the biodegradability of DOC [9–11]. During the period of DOM biodegradation, components such as protein-like, sugars, and organic acids were broken down into substances like fulvic acid and humic acid [12]. Microorganisms preferentially degraded DOM molecules with smaller molecular weights [13]. Easily degradable DOM in the early stages degraded more rapidly, while later-stage refractory DOM underwent slower degradation. Indicators of HIX (humification index), BIX (biological index), and $S_R$ are commonly used for studying patterns in DOM degradation [14,15]. It was important to note that microbial metabolism significantly influenced ultraviolet and fluorescence spectra during degradation [16]. Research has demonstrated that the degradation of DOM primarily occurs within the initial seven days, accounting for 51% to 88% of the total degradation over 28 days [17]. To obtain the quality of leaf organic matter input to the ecosystem and evaluate the biodegradability of DOC, constant temperature culture experiments had to be carried out. Previous studies focused more on the study of leachate DOC of litters, while our study focused on the study of leachate DOC degradation of fresh leaves. Moreover, our study area was an urban forest park, which had different management modes and influencing factors from ordinary forests. It was thus an important supplement to studying the biodegradation of leachate DOC of fresh leaves in an urban park.

By the end of 2023, China's forest area accounted for about 51.20% of the total land area [18]. Among which urban forest parks were concentrated in cities. Under the influence of urbanization, forest parks were built by planting trees, shrubs and grasses, and attention had to be paid to the contribution of leaf leachate to DOC. Wuhan city was located in the middle and lower reaches of the Yangtze River Basin, characterized by a subtropical monsoon climate with the widespread distribution of rivers and lakes, providing rich forest resources. The study of the biodegradation of dissolved organic carbon in fresh-leaf leachate from Wuhan Urban Forest Park has significant implications. It improves our understanding of carbon dynamics of urban park ecosystems, supporting sustainable urban planning. It also contributes to urban ecosystem protection, guides water quality management, and fosters community awareness for broader social impact. This research largely promotes practical actions for urban carbon management. This excessive fresh leaves and litters understandably affected soil organic matter content, water nutrient concentrations, and biogeochemical cycling. Understanding of DOM production and degradation processes from fresh tree-leaf leachates is essential in unraveling biogeochemical cycling of DOM, whilst this topic is not well understandable. This study aims to (1) reveal the concentration and spectral characteristics of DOM in leaf leachate, (2) investigate the influencing factors of DOC biodegradability at different temperatures, and (3) explore the generation and degradation process of DOM in leaves of different types of fresh trees.

## 2. Materials and Methods

### 2.1. Leaves Collection and Preparation of Leachates

Fresh leaves of 12 different tree species (including oak, loquat tree, heather tree, begonia in xifu, palm tree, fir, sassafras, camphor tree, pomelo tree, European oleander, holly guard spear and cuckoo) were collected. The selected 12 tree species could be divided into deciduous trees (oak, loquat tree, heather tree, begonia in xifu, palm trees, fir, sassafras, camphor tree, pomelo tree) and shrubs (European oleander, holly guard spear and cuckoo). All leaf samples were collected from the Moshan Scenic Area in the Donghu (located in Wuhan, Hubei Province, China). Wuhan has a subtropical monsoon climate characterized by abundant rainfall, ample sunshine, hot summers and cold winters. Moshan was situated in the center of Donghu, surrounded by water on three sides, and featured six winding peaks, covering a total area of 12 square kilometers. As an urban park, Moshan exhibited typical urbanization characteristics. It boasted abundant forest resources and bordered

the urban lake, Donghu. The Moshan, chosen as the sampling location for leaf specimens, allowed for a better representation of Urban Forest Parks. The extraction method of leachate was developed based on the previous methods [19–21]. The leaves were stripped of petioles, attached materials and mineral soil and then dried to a constant weight in a 70 °C oven. After drying, the leaves were cut into pieces smaller than 0.5 cm. One gram of the dried and cut leaves was placed in a large beaker, and 3000 mL of ultrapure water was added in a 1:3000 ratio of leaves to water. The mixture was sealed with plastic wrap and soaked in the dark for 36 h. The supernatant was filtered through a 0.22 μm Millipore filter to obtain the leachate.

## 2.2. Incubation Experiment of BDOC

At the sampling point (Moshan), 40 g fresh soil was added to 1000 mL ultra-pure water and cultured in an incubator at 20 °C for 12 h. After the incubation, the microbial inoculum was obtained by filtering through a 0.7 μm pore size filter [22]. In total, 180 mL of leachate was taken in a 250 mL clean glass culture bottle [23]. Two culture bottles were prepared for each type of tree-leaf leachate, and 5 mL of microbial inoculum was added to each bottle. The experiment was conducted under 20 °C and 30 °C incubation conditions, resulting in 24 culture bottles. The bottle caps were opened and shaken daily to allow air to enter the bottles and provide oxygen for microbial activities. On the 1st, 3rd, 7th, 14th, and 28th days, 30 mL of leachate was taken out for spectral and DOC measurements [16]. The rate of DOC biodegradation was assessed by the difference between the values before and after DOC biodegradation, which was then divided by the initial value to evaluate the extent of biodegradation (the degradation rate of DOC referred to the decrease in DOC concentration in leachate after 14 days relative to the initial DOC concentration and is presented as a percentage of the initial concentration).

## 2.3. Band Scanning and Spectral Measurements

The leachate was scanned using a UV (ultraviolet–visible) spectrophotometer (UV-8000), with ultrapure water as the blank. The scanning was performed in a 10 mm quartz cuvette within the 200–800 nm range, and baseline correction was carried out at 700 nm absorbance. Before sample measurement, blank correction was conducted using ultrapure water.. The absorption coefficient $\alpha(\lambda)$ represents the absorbance of the sample for UV light per unit path length [24] as follows:

$$\alpha(\lambda) = 2.303 \times A(\lambda)/i \tag{1}$$

where $A(\lambda)$ was the absorbance at wavelength $\lambda$ nm, and i was the path length of the quartz cuvette (m). Calibration was performed using the 680–700 nm wavelength range. UV spectroscopic parameters included $\alpha254$, E2/E3, E4/E6, S275–295, S350–400 and SR (Table S2 [25–28]). The concentration of CDOM was represented by $\alpha_{254}$ [25]. $E_2/E_3$ ($\alpha_{250}/\alpha_{365}$) has a significant negative correlation with DOM molecular weight [26], while $E_4/E_6$ ($\alpha_{465}/\alpha_{665}$) has a significant negative correlation with the degree of aromatic carbon ring polymerization in DOM [27]. $S_{275–295}$ and $S_{350–400}$ were spectral slopes, and $S_R$ was the ratio of $S_{275–295}$ to $S_{350–400}$ and was used to indicate the source of DOM [28].

Three-dimensional fluorescence spectra were measured using the Shimadzu RF-6000 fluorescence spectrophotometer as follows: scan speed: 1200 nm/min; scan range: Ex: 200–450 nm; Em: 250–600 nm; excitation slit width: 5 nm; and emission slit width: 1 nm. Ultrapure water was used as the EEM (excitation–emission matrix) blank to subtract Raman and Rayleigh scattering. The fluorescence intensity integral value at an Ex of 350 nm and an Em of 371–428 nm was used as the correction factor [29]. In this study, FRL (frontal Rayleigh line), BIX and HIX were used to represent the spectral characteristics of DOM. Fluorescence parameters included FRL, BIX and HIX. (Table S2 [30–32]). FRL was the ratio of fluorescence intensity at an Ex of 310 nm and an Em of 380 nm to the maximum fluorescence intensity between an Ex of 420 and 435 nm, representing the proportion of newly generated DOM in the total amount [30]. HIX was the ratio of the integral value in the Em range of 435–480 nm

to the sum of integral values in the Em range of 300–345 nm and Em range of 435–480 nm and was used to indicate the degree of humification. HIX > 0.9 indicated strong humic characteristics, while HIX < 0.8 indicated weak humic characteristics [31]. BIX was the ratio of fluorescence intensities at Em of 380 nm and 430 nm when Ex was 310 nm and was used to characterize the contribution of self-derived components in DOM. BIX > 1 indicated a strong self-derived feature [32].

### 2.4. Dreem-PARAFAC Analysis

Dreem-PARAFAC was performed using Matlab2021a. The Dreem toolbox was used for blank correction, Raman scattering correction and Rayleigh scattering correction of the water samples. All EEM data were combined to form a three-dimensional matrix for parallel factor analysis. Outliers were removed, and core consistency diagnostics were conducted using the squared excitation and emission spectra errors to assess the model components initially. Finally, a cross-validation test was performed to determine the model components. BIX, HIX, FRL and the components' fluorescence intensities were also calculated. There was no correlation between humification and absorption.

### 2.5. Water Quality Parameter Determination

Before measurement, the water samples were filtered using a Syringe Filter (Labfil, Chain) with a pore size of 0.22 μm. The pH was measured using a CyberScan PCD 650 multi-parameter water quality analyzer (Thermo Fisher Eutech, Singapore). The EC (electrical conductivity) was measured using the same CyberScan PCD 650 analyzer (Thermo Fisher Eutech, Singapore). DOC and TDN (Total Dissolved Nitrogen) were determined using a Multi N/C 2100S analyzer (Jena, Germany). TDP (Total Dissolved Phosphorus) was determined by ammonium molybdate spectrophotometry [33]. By setting the concentration gradient, the model of absorbance of the solution containing different TDP concentrations at 700 nm was developed, and then the TDP concentration in samples was obtained based on the model above.

### 2.6. Plotting and Analysis

SPSS26 was used for normality testing, and the Mann–Whitney U test was used to analyze the difference between data that did not conform to normality (differences in DOC leaching and degradation of deciduous trees and shrubs, and differences in declination and stabilization of DOC at different temperatures). The leaching rate, degradation rate and stable DOC content of DOC were calculated to evaluate the leaching and degradation of DOC in fresh leaf drench solution. The attenuation coefficient was obtained by fitting the degradation trend of DOC with the kinetic model. The data were log transformed, and then Pearson correlation analysis was carried out to evaluate the effects of water quality parameters, stoichiometric ratio (DOC, TDN, TDP), UV parameters, fluorescence parameters and leaching rate on the degradation rate of DOC by correlation coefficients. Water quality parameters and stoichiometric ratio (DOC, TDN, TDP), as well as the change curves of DOC, fluorescence parameters, UV parameters and fluorescence components, were drawn using origin2021.

## 3. Results

### 3.1. Water Quality Parameters

The initial leachate pH ranged from 5.52 (heather tree) to 6.52 (oak) with a total average of $5.95 \pm 0.27$. At the end of the 28-day incubation, the pH uniformly increased, with some leachate having a pH above 7. At 20 °C incubation, the pH ranged between 6.39 (cuckoo) and 7.38 (loquat tree) ($6.75 \pm 0.30$), and at 30 °C, the pH ranged from 6.58 (cuckoo) to 7.17 (holly guard spear) ($6.78 \pm 0.19$) (Figure S1).

The initial leachate EC ranged from 9.97 (fir) to 31.06 (oak) μS/cm (mean: $16.51 \pm 5.82$ μS/cm). After incubation, EC values exhibited an increasing trend overall. At

20 °C, the EC ranged from 11.69 (fir) to 42.57 (oak) μS/cm (20.78 ± 8.24 μS/cm), and at 30 °C, the EC ranged from 10.05 (fir) to 43.85 (oak) μS/cm (19.52 ± 8.86 μS/cm) (Figure S1).

The initial TDN and TDP concentrations in the leaf leachates were highly variable, and both of them showed the highest concentrations observed in the oak leachate (5.74 mg/L for TDN, 1.47 mg/L for TDP). The range of changes in the initial ratio of TDN to TDP in the leachate was relatively large (the average value was 7.91 ± 4.20). Among them, the maximum ratio was for fir (14.45), and the minimum ratio was for begonia in xifu (2.14) (Figure S2).

### 3.2. DOC Quantity and BDOC in Leaf Leachates

The initial DOC concentration varied between 17.84 (fir) and 39.46 (pomelo tree) mg/L, with a total mean of 28.74 ± 7.55 mg/L (Figure 1). The DOC yield per gram of leaf was 10.43%, 6.56%, 11.24%, 9.32%, 5.97%, 5.35%, 6.09%, 7.45%, 11.84%, 9.26%, 10.91% and 9.04% for the 12 tree species (oak, loquat tree, heather tree, begonia in xifu, palm tree, fir, sassafras, camphor tree, pomelo tree, European oleander, holly guard spear and cuckoo, respectively), with an average yield of 8.62 ± 2.27%. During the 28-day incubation period, the BDOC was fast over the first seven days, followed by a relatively stable trend afterward. The BDOC (20 °C) ranged from 41.09% (fir) to 83.08% (oak) with a total average of 55.29 ± 11.86%, and the BDOC (30 °C) varied from 40.13% (fir) to 75.60% (oak), with a total average of 51.90 ± 10.74%. The range of changes in the initial ratio between DOC and TDN in the leachate was relatively large (the average value was 19.12 ± 7.21). Among them, the maximum ratio was for cuckoo (33.17), and the minimum ratio was for oak (6.05) (Figure S2). The range of the initial ratio of DOC to TDP in the leachate was relatively large (the average value was 147.35 ± 86.89). Among them, the maximum ratio was obtained for the camphor tree (283.11), and the minimum ratio was for oak (23.63) (Figure S2). The average leaf DOC leaching amount of deciduous trees (81.10 ± 24.72) was lower than that of shrubs (96.70 ± 11.55), but there was no significant difference (*p* > 0.05).

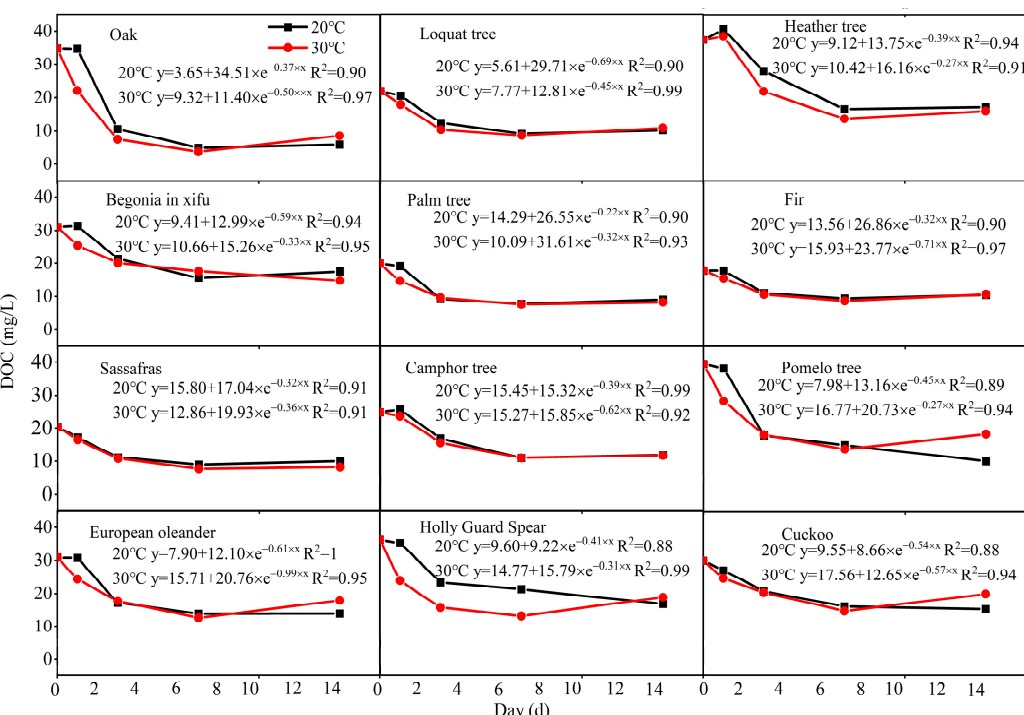

**Figure 1.** DOC changes over time during incubation process.

The DOC degradation rate ranged from 41.09% (fir) to 74.15% (pomelo tree) with a total average of 55.29 ± 11.86% at 20 °C, but it ranged from 33.60% (cuckoo) to 75.60% (oak) with a total mean of 51.90 ± 51.74% at 30 °C. The stable DOC content ranged from 17.64 mg/L

(oak) to 52.50 mg/L (begonia in xifu) with a total average of $37.20 \pm 11.18$ mg/L at 20 °C, but it ranged from 24.93 mg/L (sassafras) to 60.00 mg/L (cuckoo) with a total mean of $41.09 \pm 13.36$ mg/L at 30 °C. Deciduous trees and shrubs had no significant differences in the degradation rate and stable DOC at 20 °C, but they had significant differences at 30 °C. There was no significant difference in the degradation rate of DOC and stable DOC at the two temperatures (Table 1).

**Table 1.** Leaf leaching DOC and undegraded DOC (*p* was the statistically difference between deciduous trees (the first 9 trees) and shrubs (others), *p* was the difference between different temperatures).

| Leaf Name | Leaf DOC Loss mg/g | Degradation Rate at 20 °C % | Degradation Rate at 30 °C % | Stable DOC at 20 °C mg/g | Stable DOC at 30 °C mg/g |
|---|---|---|---|---|---|
| Oak | 104.25 | 83.08 | 75.60 | 17.64 | 25.44 |
| Loquat tree | 65.61 | 53.68 | 50.16 | 30.39 | 32.7 |
| Heather tree | 112.38 | 53.76 | 57.07 | 51.96 | 48.24 |
| Begonia in xifu | 93.15 | 43.64 | 52.59 | 52.50 | 44.16 |
| Palm tree | 59.73 | 54.14 | 58.11 | 27.39 | 25.02 |
| Fir | 53.52 | 41.09 | 40.13 | 31.53 | 32.04 |
| Sassafras | 60.93 | 50.27 | 59.08 | 30.3 | 24.93 |
| Camphor tree | 74.46 | 51.97 | 52.74 | 35.76 | 35.19 |
| Pomelo tree | 118.38 | 74.15 | 53.47 | 30.6 | 55.08 |
| European oleander | 92.64 | 54.73 | 42.1 | 41.94 | 53.64 |
| Holly Guard Spear | 109.05 | 53.78 | 48.12 | 50.4 | 56.58 |
| Cuckoo | 90.36 | 49.14 | 33.60 | 45.96 | 60.00 |
| Mean value | $86.21 \pm 22.65$ | $55.29 \pm 11.86$ | $51.90 \pm 10.74$ | $37.20 \pm 11.18$ | $41.09 \pm 13.36$ |
| *p* | 0.13 | 0.78 | 0.03 * | 0.17 | 0.02 * |
| *p* | - | 0.56 | | 0.42 | |

\* $p < 0.05$

The BDOC of the leaching solution was determined by a first-order kinetic model. By fitting the concentration changes over a period of 14 days, the decay coefficients (k values) for oak, loquat tree, heather tree, begonia in xifu, palm tree, fir, sassafras, camphor tree, pomelo tree, European oleander, holly guard spear and cuckoo at temperatures of 20 °C and 30 °C were obtained, as shown in Figure 1. The range of decay coefficients was from 0.22 to 0.99 (Figure 1).

### 3.3. Fluorescent and UV Parameters

Throughout the period of incubation, the FRL and BIX showed an overall upward trend, irrespective of the plant leaves and incubation temperature. The initial FRL ranged from 0.18 (oak) to 1.14 (pomelo tree) with a total average of $0.39 \pm 0.26$. At the end of incubation, the FRL ranged from 0.18 (cuckoo) to 1.50 (pomelo tree) with a total mean of $0.62 \pm 0.34$ at 20 °C, but it ranged from 0.23 (cuckoo) to 11.47 (pomelo tree) with a total mean of $0.72 \pm 0.37$ at 30 °C. The initial BIX ranged from 0.18 (oak) to 1.37 (pomelo tree) with a total average of $0.42 \pm 0.32$. At the end of incubation, the BIX ranged from 0.19 (cuckoo) to 2.01 (pomelo tree) with a total mean of $0.69 \pm 0.47$ at 20 °C, but it ranged from 0.24 (cuckoo) to 1.94 (pomelo tree) with a total mean of $0.78 \pm 0.47$ at 30 °C. The initial HIX ranged from 0.13 (cuckoo) to 0.77 (pomelo tree) with a total average of $0.34 \pm 0.20$. At the end of incubation, the HIX ranged from 0.24 (fir) to 4.17 (oak) with a total mean of $1.19 \pm 1.31$ at 20 °C, but it ranged from 0.28 (camphor tree) to 3.09 (oak) with a total mean of $1.27 \pm 0.94$ at 30 °C (Figure 2).

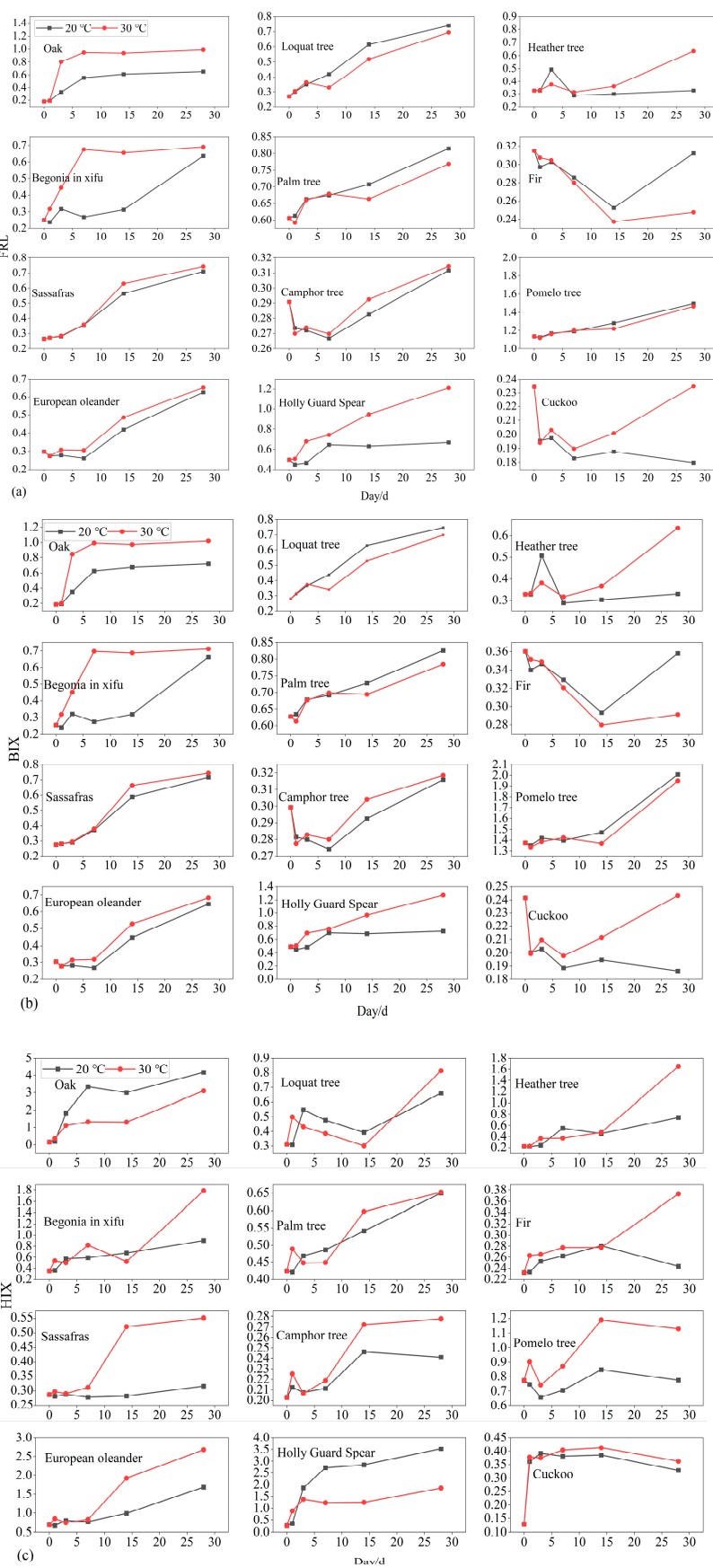

**Figure 2.** Fluorescence parameters FRL (**a**), BIX (**b**) and HIX (**c**),over time during the process of incubation.

The initial $\alpha_{254}$ ranged from 13.77 m$^{-1}$ (sassafras) to 139.28 m$^{-1}$ (oak) with a total average of 50.30 ± 34.69 m$^{-1}$. At the end of incubation, the $\alpha_{254}$ ranged from 10.73 (sassafras) to 36.86 m$^{-1}$ (begonia in xifu) with a total mean of 22.22 ± 8.52 m$^{-1}$ at 20 °C, but it ranged from 10.59 m$^{-1}$ (sassafras) to 39.88 m$^{-1}$ (begonia in xifu) with a total mean of 22.89 ± 8.45 m$^{-1}$ at 30 °C. The initial $E_2/E_3$ ranged from 1.43 (oak) to 7.89 (fir) with a total average of 4.12 ± 1.88. At the end of incubation, the $E_2/E_3$ ranged from 1.39 (begonia in xifu) to 4.91 (loquat tree) with a total mean of 3.65 ± 0.97 at 20 °C, but it ranged from 1.42 (begonia in xifu) to 4.13 (heather tree) with a total mean of 3.50 ± 0.76 at 30 °C. The initial $E_4/E_6$ ranged from 1.75 (holly guard spear) to 75.50 (begonia in xifu) with a total average of 8.93 ± 20.98. At the end of incubation, the $E_4/E_6$ ranged from 2.57 (palm trees) to 48.67 (begonia in xifu) with a total mean of 9.79 ± 12.94 at 20 °C, but it ranged from 3.21 (loquat tree) to 39.82 (begonia in xifu) with a total mean of 9.88 ± 9.93 at 30 °C. The initial $S_R$ ranged from 0 (loquat tree, heather tree and begonia in xifu) to 1.26 (fir) with a total average of 0.46 ± 0.46. At the end of incubation, the $S_R$ ranged from 0 (begonia in xifu) to 2.25 (fir) with a total mean of 0.71 ± 0.66 at 20 °C, but it ranged from 0 (begonia in xifu) to 1.66 (fir) with a total mean of 0.75 ± 0.50 at 30 °C (Figure 3).

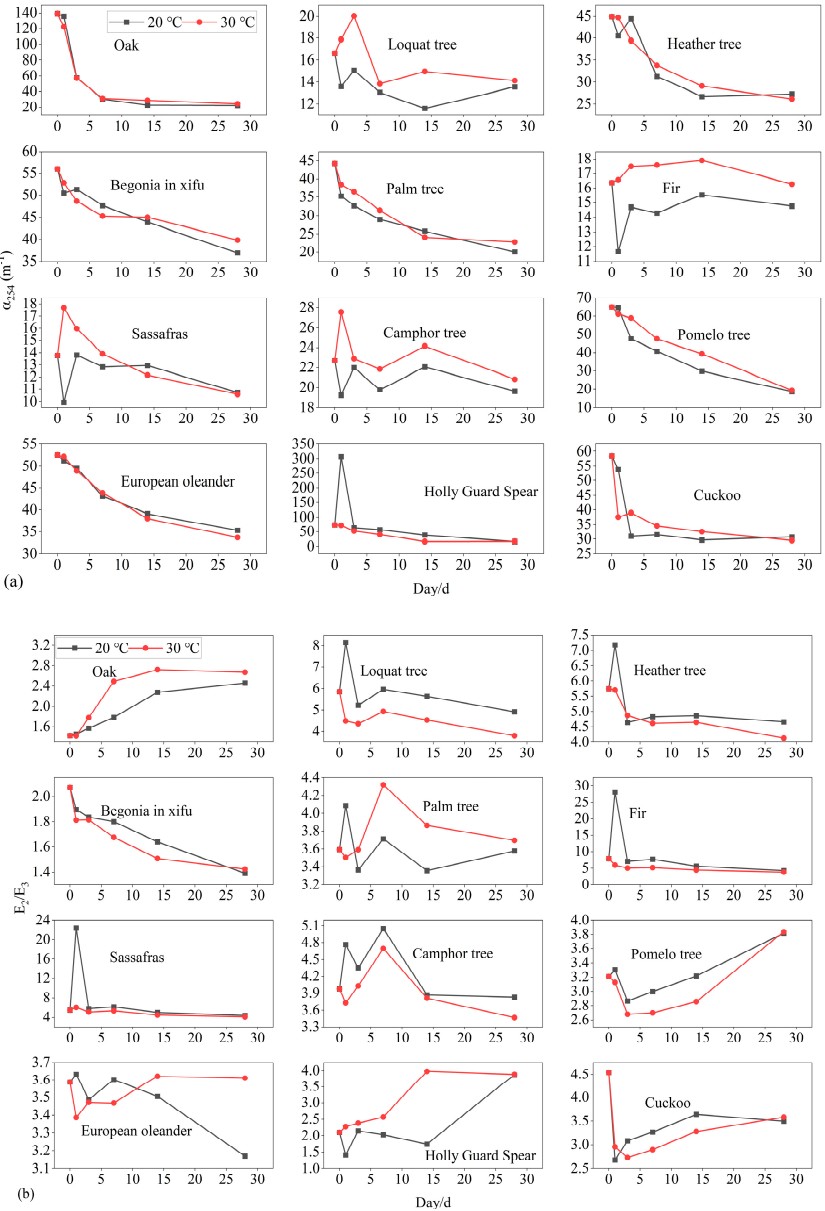

**Figure 3.** *Cont.*

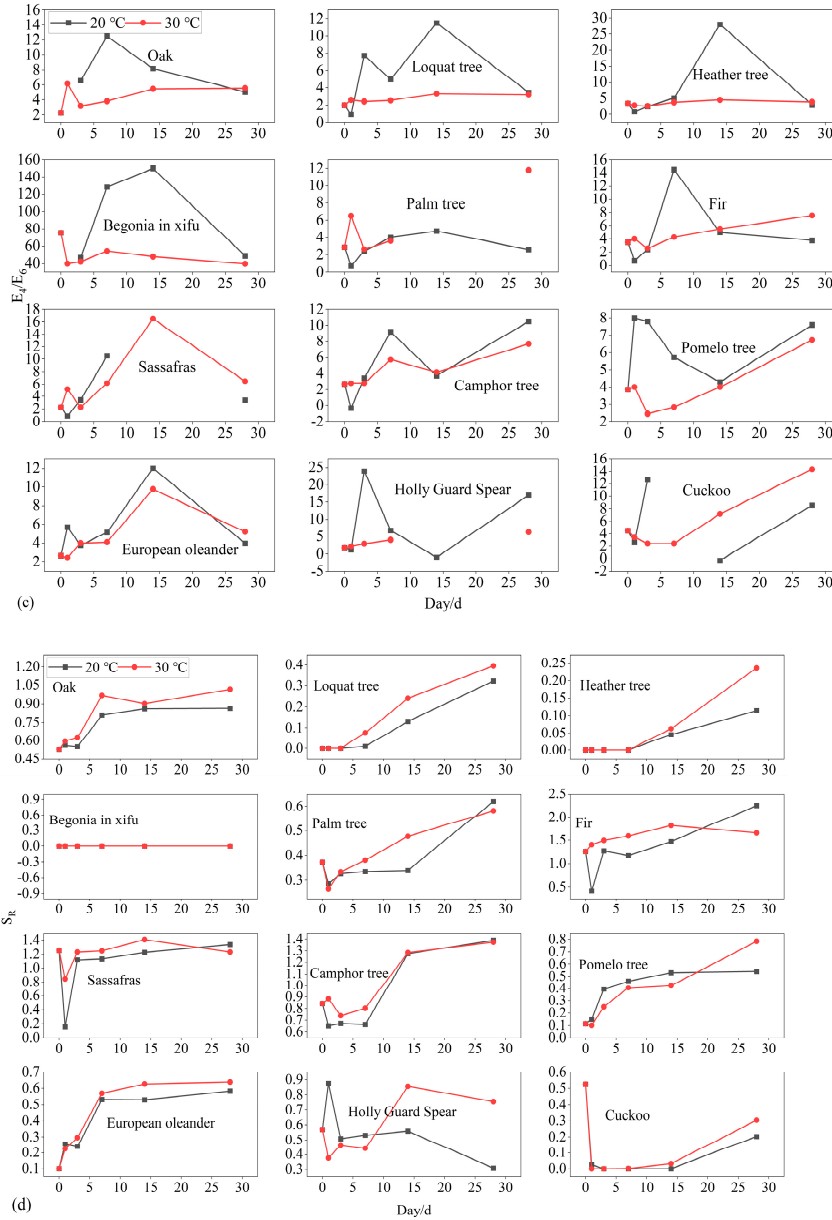

**Figure 3.** UV parameters $\alpha_{254}$ (**a**), $E_2/E_3$ (**b**), $E_4/E_6$ (**c**) and $S_R$ (**d**) during incubation process.

### 3.4. Fluorescent Components during the Degradation Process

After subtracting ultrapure water blank and eliminating Raman and Rayleigh scattering, the obtained images depicted the spectral changes of dissolved organic matter over a period of 28 days, as well as fluorescence component maps. It was observed that throughout this entire process, there was a transition in color from yellow to blue in the DOM spectrum (Figures S3–S26). The culture process of the oak tree solution was taken as an example, and there were distinct variations in four fluorescence components: The fluorescence proportion of component C1 (protein-like of microbial origin) exhibited an upward trend at an ambient temperature of 30 °C until it reached stability on day seven; the fluorescence proportion of component C2 (tryptophan-like) initially decreased before showing an upward trend with its lowest proportion observed on day three; the fluorescence proportion of component C3 (humus-like substance) initially decreased before showing an upward trend; while the fluorescence proportion of component C4 (tyrosine-like) consistently displayed a downward tendency throughout this period. In an environment of 20 °C, the fluorescence proportion of component C1 (protein-like of microbial origin) showed an increasing trend and reached stability on the 7th day. The fluorescence proportion of component C2 (tryptophan-like)

exhibited a decreasing trend and also reached stability on the 7th day. The fluorescence proportion of component C3 (humus like substance) displayed a decreasing trend and reached stability on the 7th day as well. The fluorescence proportion of component C4 (tyrosine-like), on the other hand, showed a continuous decreasing trend (Figure 4 and Table S3).

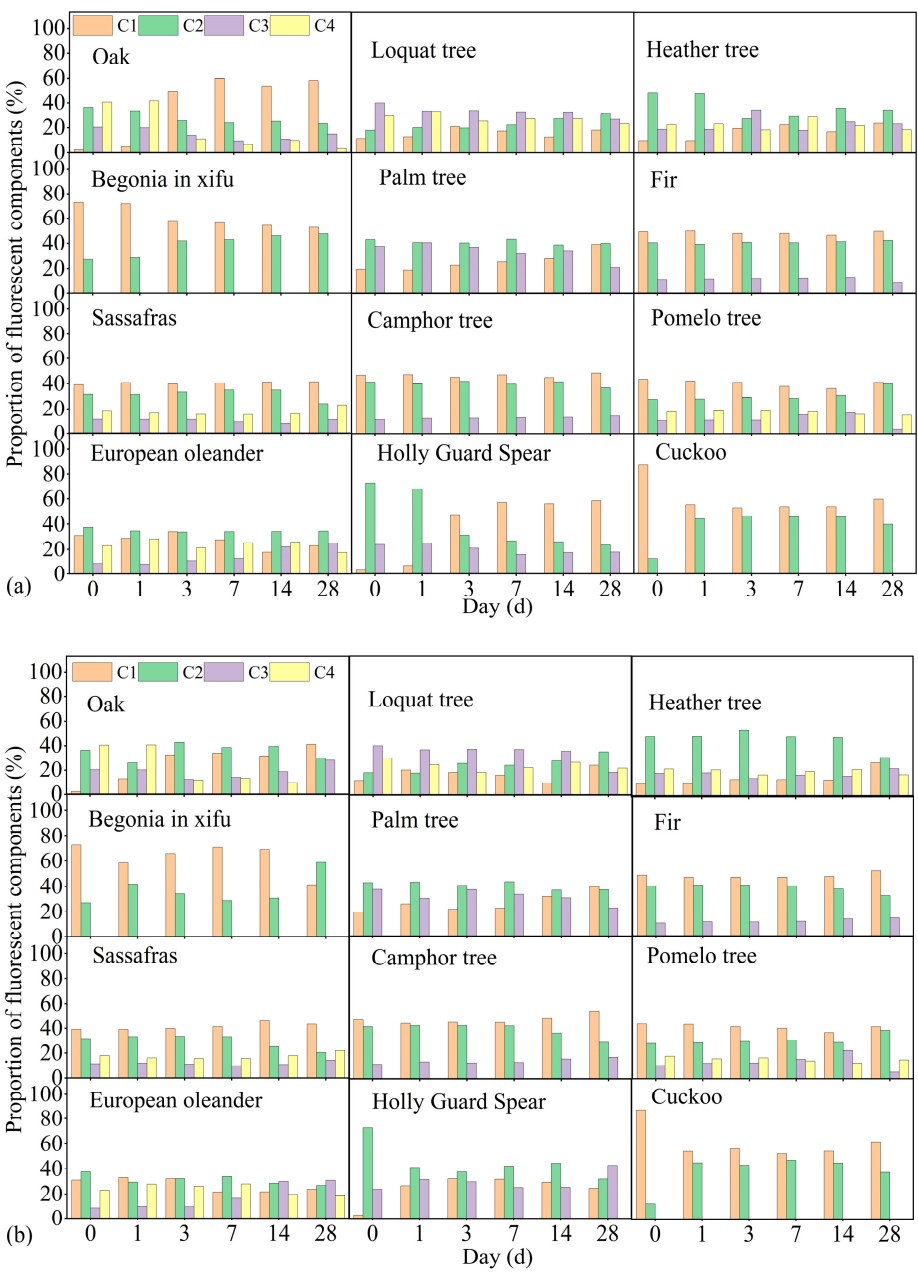

**Figure 4.** The proportion of fluorescent components in the degradation process of leaching solution at 20 °C (**a**) and 30 °C (**b**).

The tryptophan-like and protein-like substances were obtained from the changes in fluorescence intensity during the process of component degradation, and the degradation of tryptophan-like and protein-like substances was continuous. The degradation trend of the humus-like substance in most eluvial solutions was weak. The humic-like acid of the leaching solution showed a decreasing trend, and some of them showed a trend of first increasing and then decreasing (Figures S27–S32).

## 4. Discussion

### 4.1. Differences in DOC Leaching and Degradation among Different Tree Leaves

The findings revealed distinct variations in leached DOC content across diverse leaf types. Despite the influence of heightened temperatures, the degradation rate of DOC in leaf leaching solutions over 14 days showed no significant disparity. Under the temperature gradient set in the experiment, there was no significant difference in the degradation rate of the leachate of the same leaves. Therefore, we speculated that the influence of temperature on the degradation rate might not dominate the influencing factors, but the composition of leaves or other factors were the main reasons affecting the degradation rate. Certain leachates, however, exhibited discrepant DOC concentrations. Hobbie et al. [2] reported that differing leaching rates of DOC from litter among various tree species could substantiate these observations. Leaves predominantly harbored organic compounds like carbohydrates, proteins, proteinoids, lipids, chlorophyll, and nucleic acids. The chlorophyll content varied depending on leaf types, with shrubs, receiving less sunlight than deciduous trees, exhibiting higher chlorophyll content in adaptation to their light conditions, thereby generating more chlorophyll [34]. Nevertheless, diversity of the forest park needs to be cautiously included.

The study showed no significant differences in the degradation rate and stable DOC of leachate under different temperatures ($p > 0.05$). Analysis of degradation patterns indicated that among diverse tree species, fir and cuckoo maintained the highest stable DOC content under leaching conditions. This suggests that the cultivation of fir and cuckoo in urban forest parks could potentially enhance carbon sequestration and bolster the carbon sink capacity in these forested areas.

### 4.2. Kinetic Modeling of the Degradation Process of DOM

The degradation of DOM followed a first-order kinetic model [35], and thus, the equation $y = a + b \times e^{(-k \times x)}$ can be used to model the changes in DOC degradation [36]. The $R^2$ values (0.88–1.00) obtained from kinetic fitting were in line with those reported for DOC photocatalyzed degradation (0.96–1.00) [35]. The European oleander exhibited the highest decay coefficient. This suggested the potential of oleander to augment carbon cycling within the ecosystem. However, it was imperative to acknowledge certain limitations. The attenuation coefficients were derived within controlled indoor conditions, possibly introducing errors into the experimental setup [37]. In natural environments, diverse factors like pH levels, light intensity, temperature fluctuations and the presence of inorganic substances wield substantial influence over microbial activities, thus impacting the degradation of DOC [38,39]. Our experimental design overlooked the influence of these external conditions, indicating potential uncertainty in the estimated attenuation coefficients.

The k value obtained exhibited no notable correlation with water quality parameters, UV–vis parameters, fluorescence parameters, fluorescence proportion and degradation rate (Table 2). This supported that the degradation process of DOC in leaf leaching solution followed a first-order kinetic model, and the influencing factors and mechanisms of the attenuation coefficient remained unclear and required further study.

**Table 2.** Correlation analysis of DOC leaching rate, DOC degradation rate at 20 °C and degradation rate at 30 °C with water quality parameters, UV–vis parameters and fluorescence parameters.

| Project | pH | EC | TDP | TDN | $\alpha_{254}$ | $E_2/E_3$ | $E_4/E_6$ | TDN/TDP | DOC/TDN | DOC/TDP | $S_R$ | FRI | BIX | HIX | DOC Leaching Rate |
|---|---|---|---|---|---|---|---|---|---|---|---|---|---|---|---|
| DOC degradation rate at 20 °C | 0.39 | 0.86 ** | 0.54 | 0.86 ** | 0.61 * | −0.55 | −0.37 | −0.02 | −0.70 * | −0.43 | −0.50 | 0.57 | 0.54 | −0.58 * | 0.54 |
| DOC degradation rate at 30 °C | 0.19 | 0.53 | 0.60 * | 0.63 * | 0.22 | −0.46 | −0.08 | −0.30 | −0.67 * | −0.66 * | 0.06 | 0.18 | 0.21 | −0.53 | 0.14 |
| k at 20 °C | 0.21 | 0.12 | 0.22 | −0.20 | 0.09 | −0.08 | 0.30 | −0.49 | 0.48 | −0.13 | −0.50 | 0.11 | 0.31 | −0.03 | 0.38 |
| k at 30 °C | 0.56 | −0.34 | −0.35 | −0.28 | −0.18 | 0.20 | −0.18 | 0.25 | 0.14 | 0.30 | 0.00 | −0.19 | −0.28 | 0.14 | −0.31 |

$*\ p < 0.05$, $**\ p < 0.01$.

### 4.3. Main Factors Influencing DOM Degradation

We revealed that the degradation rate of DOC for leaf leaching solutions was notably influenced by the nutrient concentration and the content of DOM humification, and the degradation rate was different with temperature. Temperature mediated the activity of microbial populations, thereby influencing the degradation process of DOM [40–42]. We found that under 20 °C conditions, the degradation rate of DOC was significantly positively correlated with EC and TDN (r = 0.86); the DOC/TDN ratio showed a significant negative correlation (−0.70) (Table 2). EC reflected the solute/salinity in the environment and increased salinity-promoted microbial decomposition rates under low salinity conditions [43]. Furthermore, inorganic nitrogen availability also controlled the biodegradation of DOC by limiting microbial growth [44]. Thus, EC and TDN showed a strong positive correlation with the degradation rate. On the other hand, there was a significant negative correlation between the HIX and the degradation rate (r = −0.58, $p < 0.05$) (Table 2). It was understandably reasonable that higher humidification means more recalcitrant organic substance. As DOM degraded, the H/C ratio decreased, and the humification degree increased.

Under the condition of 30 °C, the degradation rate of DOC showed a significant positive correlation with TDP (r = 0.60, $p < 0.05$) and TDN (r = 0.63, $p < 0.05$). Both DOC/TDN and DOC/TDP ratios, however, showed a negative correlation, with correlation coefficients of −0.67 and −0.66, respectively (Table 2). It was reported that bacterial metabolic activity depended on the availability of phosphorus, and thus, bacteria had vigorous metabolism and can easily degrade DOC under conditions of high TDP concentration [45], showing a positive correlation. However, our exploration did not encompass the impact of DOM molecule composition on the degradation rate. This aspect remains an area for future investigation and will be a focus of our forthcoming studies. Understanding the contribution of DOM molecule composition to the degradation rate will offer a more comprehensive insight into the intricate processes governing DOC dynamics within leaf leaching solutions. We took the degradation rate as the dependent variable, and the parameters that significantly correlated with the degradation rate were the independent variables, and the following regression models were obtained: $y_{\text{degradation rate at 20°C}} = 0.541x_{\text{EC}} + 1.087$ (r = 0.87, $p < 0.01$), $y_{\text{degradation rate at 20°C}} = 0.309x_{\text{TDN}} + 1.67$ (r = 0.86, $p < 0.01$), $y_{\text{degradation rate at 20°C}} = -0.302x_{\text{DOC/TDN}} + 2.111$ (r = 0.87, $p < 0.05$), $y_{\text{degradation rate at 20°C}} = -0.23x_{\text{HIX}} + 1.669$ (r = 0.58, $p < 0.05$), $y_{\text{degradation rate at 30°C}} = 0.142x_{\text{TDP}} + 1.796$ (r = 0.60, $p < 0.05$), $y_{\text{degradation rate at 30°C}} = 0.243x_{\text{TDN}} + 1.659$ (r = 0.63, $p < 0.05$), $y_{\text{degradation rate at 30°C}} = -0.184X_{\text{DOC/TDP}} + 2.08$ (r = 0.66, $p < 0.05$), and $y_{\text{degradation rate at 30°C}} = -0.312X_{\text{DOC/TDN}} + 2.08$ (r = 0.67, $p < 0.05$).

### 4.4. Changes in Fluorescent Components during the Degradation Process

The biodegradability of DOM was mainly caused by the instability of carbohydrates and protein-like substances in the fresh biomass of leaves [46]. As humic-like acid fluorescence was generated in the microbial degradation process of DOM, the protein-like fluorescence was reduced [47], while humic-like acid had biological resistance and was not easy to biodegrade [48]. Even under anaerobic conditions, DOM degradation involved the production of humic-like acid and degradation of tyrosine-like substances [49]. In the process of DOM degradation, the humus-like components did not change and were stable [50]. Our experimental results underscore that microbial degradation of leaf leachate predominantly involves the breakdown of protein-like components with a minor proportion of humic-like acid. This highlights the selective degradation of specific DOM constituents during microbial processes, indicating a preference for degrading protein-like compounds rather than humic-like substances. We found that HIX had a significant negative correlation with the degradation rate of DOC, which confirmed that the microbial degradation of DOC was more inclined to degrade protein-like compounds than humic-like substances.

## 5. Conclusions

Different tree species had different leaching rates of carbon chemo-diversity and nutrients. Thus, tree species are crucial for carbon sinks and carbon losses to the aquatic environment. The fresh leaf-leached DOC quality and biodegradability, as well as their influencing factors from 12 major tree species, including deciduous trees and shrubs, are unraveled. The results showed that the leachate DOC degradation rate of fir (compared in nine middle trees) and cuckoo (compared in three middle shrubs) leaves was lower. The DOC of the tree species with low degradation rate was more stable and could play a role in carbon sequestration. Their selection as the main tree species for urban green land would improve the carbon sequestration ability of urban forests. Oak was more biodegradable and contributed to the efficient carbon cycling of woodlands, thus being better assimilated into forest ecosystems.

**Supplementary Materials:** The following supporting information can be downloaded at: https://www.mdpi.com/article/10.3390/w16040558/s1, Figure S1: pH (a) and EC (b) changes on 0d and 28d incubation.; Figure S2: TDN and TDP concentrations (a), TDN/TDP (b), DOC/TDN (c) and DOC/TDP (d) in fresh-leaf leachate; Figure S3: Oak tree-leachate DOM degradation process. Figure S4: Four components of fluorescent substances in oak-tree leachate. Figure S5: Loquat-tree leachate DOM degradation process. Figure S6: Four components of fluorescent substances in loquat-tree leachate. Figure S7: DOM degradation process of heather-tree leachate. Figure S8: Three components of fluorescent substances in heather-tree leachate. Figure S9: DOM degradation process of begonia in xifu-tree leachate. Figure S10: Four components of fluorescent substances in begonia in xifu-tree leachate. Figure S11: DOM degradation process of palm-tree leachate. Figure S12: Two components of fluorescent substances in palm-tree leachate. Figure S13: DOM degradation process of fir-tree leachate. Figure S14: Three components of fluorescent substances in fir-tree leachate. Figure S15: DOM degradation process of sassafras leachate. Figure S16: Four components of fluorescent substances in sassafras leachate. Figure S17: DOM degradation process of camphor-tree leachate. Figure S18: Three components of fluorescent substances in camphor-tree leachate. Figure S19: DOM degradation process of pomelo-tree leachate. Figure S20: Four components of fluorescent substances in pomelo-tree leachate. Figure S21: DOM degradation process of European-oleander leachate. Figure S22: Four components of fluorescent substances in European-oleander leachate. Figure S23: DOM degradation process of holly guard-spear leachate. Figure S24: Four components of fluorescent substances in holly guard-spear leachate. Figure S25: DOM degradation process of cuckoo leachate. Figure S26: Four components of fluorescent substances in cuckoo leachate. Figure S27: Tyrosine-like fluorescence intensity change process. Figure S28: Tryptophan-like fluorescence intensity change process. Figure S29: Protein-like fluorescence intensity change process. Figure S30: Humic-like substance fluorescence intensity change process. Figure S31: Humic-like acid fluorescence intensity change process. Figure S32: The change of fluorescence intensity of microbial source protein-like and amino-like acid. Table S1: Abbreviation and symbol mapping table. Table S2: Parameter significance. Table S3: Fluorescent substances in the fresh-leaf-leached DOC. References [25–28,30–32] are cited in the supplementary materials.

**Author Contributions:** Conceptualization, X.T. and S.L.; methodology, X.T. and S.L.; software, X.T.; validation, S.L.; formal analysis, X.T.; investigation, X.T. and S.L.; resources, S.L.; data curation, X.T.; writing—original draft preparation, X.T. and S.L.; writing—review and editing, S.L.; visualization, S.L.; supervision, S.L.; project administration, S.L. All authors have read and agreed to the published version of the manuscript.

**Funding:** The study was financially supported by the funding from Wuhan Institute of Technology to Dr. S.L. (21QD02) and Wuhan Institute of Technology Graduate Education Innovation Fund Project (CX2023158).

**Data Availability Statement:** The data presented in this study are available on request from the corresponding author.

**Conflicts of Interest:** The authors declare no conflicts of interest.

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
