# Peer review of "Quality and Biodegradation Process of Dissolved Organic Carbon in Typical Fresh-Leaf Leachate in the Wuhan Urban Forest Park"

_water, doi:10.3390/w16040558_

Round 1

Reviewer 1 Report

Comments and Suggestions for Authors

Key topic of the manuscript is carbon sequestration capacity of urban forests by investigating fresh-leaf leachate biodegradation.  However, I did not find their relationship after I read the manuscript carefully.  Therefore, I did not recommend it for publication in Water.

1. What is relationship between the leachate biodegradation and  carbon sequestration capacity?

2. Experimental design for leachate biodegradation was unreasonable, because the culture bottles are much different from actual environment, and results will be hard to convince.

3.  Figures 1 and 2 are insignificant, as they did not give special explanation for whole manuscript.

4. Kinetic models of BDOC in fresh leaf leachate did not give any regularity for carbon  suggest carbon sequestration.

5. Authors should give some components of each leaf, such as organic acids etc., and the DOC is insufficient.

6. Figure 3 is too terrible to differentiate difference among the leaves for their fluorescence parameters clearly.

Reviewer 2 Report

Comments and Suggestions for Authors

V

Review Report

Paper Title: Quality and biodegradation process of dissolved organic carbon  in typical fresh-leaf leachate in the Wuhan Urban Forest Park

Authors: Xiaokang Tian & Siyue Li

Paper Nr: 2817275

1.    General Remarks:

In this research paper, the leaching and biodegradation potential of dissolved organic carbon (so-called “BDOC”, from which I understand “biodegradable DOC”) in leaf leachates from typical fresh leaves in the Wuhan Urban Forest Parks, Central China, was examined for 12 major tree species. Biodegradation was followed with an incubation procedure for 28 days. The 28-days incubation experiment was conducted at two temperatures (T = 20 °C and 30 °C) to study the effect of temperaure on the organic carbon abatement (ultimate oxidation – mineralization). The DOC data could be fitted to pseudo-first order kinetics. The authors indicated that there was no significant difference in the degradation rate of the leaching solution at different temperatures, but the k values changed with temperature. The authors established some correlations among the followed process paramters. At T = 20 °C, the DOC abatement positively correlated with Electrical Conductivity and Total Dissolved Nitrogen (TDN), and negatively correlated with the Humification Index and the DOC/TDN ratio. At T = 30 °C, DOC abatements positively correlated with Total Dissolved Phosphorus (TDP) and TDN, and negatively correlated with the humification index, DOC/TDN and DOC/TDP ratios.

The authors claim that the elucidation of DOM production and degradation processes from fresh tree leaf leachates is important to understand the biogeochemical cycling of DOM. In the present study they mainly aim to reveal the concentration and spectral characteristics of DOM in leaf leachate, investigate major factors affecting the biodegradability of DOC at different temperatures and explore the generation and degradation of DOM in leaves indifferent types of fresh trees. In their Conclusions section, information given in the Abstract Section is repeated.

Generally speaking, this is an interesting and comprehensive study, however, its overall impact and scientific contribution seems rather unclear to me. Besides, English/writing style could be improved since some parts of the paper were quite difficult to follow. Some more spesific comments are listed below.

2.    Specific Comments:

·       Abstract: No abbreviations and symbols should be used in this section. At least not directly, since these may not be clear/known to the reader.  Also, what is BDOC standing for? Also not clear, we can guess it is the biodegradable fraction of the total DOC. So, it is not clear what abbreviation BDOC is used for (probably it is “biodegradable DOC”-?).

·       Abstract and elsewhere: Please edit “humification” to “humification index” (last sentence in this section).

·       A list of Abbreviations and Symbols would be very useful to the reader. There are numerous of them in the manuscript and at least with some of these the reader may not be familiar.

·       Introduction and elsewhere: DOM and other parameter symbols = Please first indicate the full name of the parameter and then the abbreviation that was mentioned in the first paragraph of this section fort he first time, in the second sentence ……(DOM, line 29).   

·       Experimental: All methods (analytical, instrumental) used in this work should be cited with appropriate references.

·       Band scanning and spectral measurements: The authors use specific wavelengths to interpret different aromaticity and other structural properties of organic matter that may correlate with degradability and molecular weight of DOM. However, absorbance (spectrometric) measurements give only a very general and rough estimation of the organic matter content, aromaticity and other molecular properties. Better is to support this data by direct organic carbon measurements, with TC-DOC instrumental analysis and is fractionation according to particle size and other structural features and properties of NOM-DOM.

·       Section 2.3. Please number reaction equations and formulae. The pathlength unit should be cm, if the value is one (1), not meters (pathlength is typically 1 cm for spectrophotometers).

·       Section 2.5. Water quality parameter determination: TDP (Total Dissolved Phosphorus) was measured using a spectrophotometric method; Here, more details should be given, such as the reference of the method (for instance, APHA-AWWA-WPCF, year of publication?) and calibration/working range for all parameters followed in this paper.

·       Results: In this section, the authors report (expected) parameter values for different tree leachates. The authors could have combined this section with the “Discussion” section to explain the relationship between the parameters and what conclusions could be drawn from these relationships. Combining these two sections is generally preferred, since it is easier and more practical to follow results and its discussion closely-together.

·       Results: Please explain what is meant with (DOC) degradation rate (it is given in percent). Is is “DOC removal efficiency”? Rate is about speed of the reaction. The degradation rate and degradation rate coefficients are defined in a different way and thus have different units. The unit of degradation rate is fixed and typically “concentration time-1” and that of the (degradation) rate coefficient changes with kinetic model fitting the experimental data, but typically is first- order in environmental applications (unit of k = rate coefficient = time-1)

·       Please indicate all experimental conditions in figure captions  and table headings.

·       Figures: Margins of experimental errors are missing in the figures and should be added into the data presentation of the figures. Since the authors conducted a statistical analysis for their experimental study, this should not be a problem.

·       Results:

-The kinetic rate coefficients are presented in Table 2, therefore it is not necessary to list all the calculated rate coefficients in the paragraph right before (above) this table. In this way, information is only dublicated.

-For data presentations, figures are generally speakng more useful and visually more attractive to examine experimental data. Therefore, the authors may consider transforming some of their table sets to 2-D figures (column or scatter figures).

·       Discussion:

-It is typically expected that k values (rate coefficients) increase with temperature (T) this is also in line with the Arrhenius approach. However, in this study, the obtained kinetic rate coefficients do not always follow this trend. In other words, k is not increasing with elevating the T. This should be explained herein, since it is a surprising observation.

-A more in-depth evaluation and discussion is expected for the followed environmental parameters (DOC, all fluorimetric and spectrometric data, N and P concentrations, etc.) and besides modeling the kinetics of the DOC - BDOC degradation pattern, an empirical model could also be developed for the relationship and interaction between these followed parameters.

·       Discussion of Section 4.4 The authors write here the following: “Our experimental results underscore that microbial degradation of leaf leachate predominantly involves the breakdown of protein-like components with a minor proportion of humic-like acid. This highlights the selective degradation of specific DOM constituents during microbial processes, indicating a preference for degrading of protein-like compounds rather than humic-like substances.” This is quite obvious and well-known. More specifically, that biodegradation of DOC is a microbial process and the breakdown of protein-like susbstances is easier to perform than that of humic-like substrates are known facts. Humic acid and similar substances are difficult to degrade because of complex, structural features, aromaticity and high molecular weight. Hence, it is not surprising (but rather expected) that the bio-resistance of humic-like materials will be reflected in the DOC,UV254 and fluorescence data. Considering these instrumental findings, the authors may discuss and explain here what is original/new in the present paper.

·       Conclusions: The study may provide information about what tree or leaf types are more biodegradable and contribute to an efficient Carbon Cycle in forest lands, and thus which tree/leaf types are better assimilated by the forest ecosystem an which are not to chose the right vegetation. This should be emphasized in this section.

·       Conclusions: In this section, the authors repeat the information they already gave in the “Abstract” section. Instead, a briefing of their most important and interesting findings, some recommendations for future work, lessons learnt from this experimental study could be given herein. The authors are hence encouraged to re-write and re-phrase this section and instead of writing something about their major findings they could drawn major conclusion from the present study.

·       References: This section needs some update. In the Reference List, more recent, related work should be included, most references are ok regarding the scope & concept but rather old.

My final, general opinion about this paper is that the topic, scientific approach and experimental aim & scope of the present work do not really fit to the journal’s concept so well. Besides, the paper is very specific and describes a local situation/problem about the forest ecosystem of Wuhan city. In the Introduction (last paragraph) and Conclusion Sections the main (rea) problem addressed, its global significance should also be emphasized in the revised paper.

Comments on the Quality of English Language

As stated above, some parts are difficult to read and understand. It is advisable to edit the manuscript by the assistance of a native speaker.

Reviewer 3 Report

Comments and Suggestions for Authors

The draft manuscript presents an interesting study related to the bio-degradation of dissolved organic carbon in fresh-leaf leachate specific to Wuhan Urban Forest Park. After a detailed review I have the following comments and suggestions.

1.      The novelty of the objective is not clear. There are ample number of similar scientific studies on the bio-degradation of dissolved organic carbon. I would like to know how the present work is different from the existing studies.

2.      As far as I could comprehend, the observations are specific to Wuhan Urban Forest. What motivated the authors to select this study region?

3.      Line 33 – “At present, researchers have …….”. This line is not making any sense. Try replacing At present with Till now or something else.

4.      All in all, INTRODUCTION section is too short and many important studies (recent and relevant to the present work) have not been referred to. I advise the authors to rewrite this section including all the recent and relevant scientific literature.

5.      MATERIALS AND METHODS – This section is well-written. But it is not clearly mentioned how the authors have formulated the complete methodology. For E.g. – A very specific procedure has been adopted by the authors for leaves collection and preparation of leachates. Is it a text-book methodology or based on some previous scientific studies? Kindly provide the source of the methodology described or if the authors have devised the methodology, it is my suggestion to the authors to provide the scientific basis of the overall methods.

6.      Water quality parameters chosen for the present study are pH, electrical conductivity, dissolved organic carbon, total dissolved nitrogen and total dissolved phosphorus. Is this list comprehensive? If no, what other parameters can be included in such studies?

7.      The best part of this draft is the RESULTS AND ANALYSIS section. I really enjoyed reading this section which is nicely written and the conclusions are logically drawn from the observations.

I genuinely believe that the above suggestions, if incorporated, will greatly enhance the quality of the current draft.

Comments on the Quality of English Language

Should be improved.

Round 2

Reviewer 1 Report

Comments and Suggestions for Authors

The manuscript introduced a story that leaves released nutrient which resulted in carbon cycle or potential carbon sequestration. This can direct how to plant what tree in urban zone. However, some revisions are necessary before acceptation for publication.

1. For the measure, a explanation for the relationship between humic characteristics and absorption is required in M&M.

2. I wonder how authors culture without any nutrient in the medium (line 114)?

3. Why can the trees with low DOC degradation rate improved the carbon sequestration ability of urban forest, for which some reasonable explanations are required.

Comments on the Quality of English Language

The manuscript should be polished before publication.

Reviewer 2 Report

Comments and Suggestions for Authors

The authors have addressed most comments and recommendations made by the reviewers to improve their manuscript. The instrumental procedures, model development and statistical assessment are still rather weak, however, not so critical for the quality improvement of the paper.

Comments on the Quality of English Language

The English is acceptable. Minor editing (a final round for remaining typos and language) is recommended.

Reviewer 3 Report

Comments and Suggestions for Authors

The authors have revised the manuscript. I therefore recommend it for acceptance. 

Comments on the Quality of English Language

Once proof read would be better. 

Round 3

Reviewer 1 Report

Comments and Suggestions for Authors

Authors have improve their manuscript, and it reached at the level for publication.